# An Efficient Supervised Deep Hashing Method for Image Retrieval

**DOI:** 10.3390/e24101425

**Published:** 2022-10-07

**Authors:** Abid Hussain, Heng-Chao Li, Muqadar Ali, Samad Wali, Mehboob Hussain, Amir Rehman

**Affiliations:** 1School of Computing and Artificial Intelligence, Southwest Jiao Tong University, Chengdu 611731, China; 2Department of Mathematics, Namal Institute, Mianwali 42250, Pakistan

**Keywords:** deep learning, deep supervised hashing, convolutional neural network, image retrieval

## Abstract

In recent years, searching and retrieving relevant images from large databases has become an emerging challenge for the researcher. Hashing methods that mapped raw data into a short binary code have attracted increasing attention from the researcher. Most existing hashing approaches map samples to a binary vector via a single linear projection, which restricts the flexibility of those methods and leads to optimization problems. We introduce a CNN-based hashing method that uses multiple nonlinear projections to produce additional short-bit binary code to tackle this issue. Further, an end-to-end hashing system is accomplished using a convolutional neural network. Also, we design a loss function that aims to maintain the similarity between images and minimize the quantization error by providing a uniform distribution of the hash bits to illustrate the proposed technique’s effectiveness and significance. Extensive experiments conducted on various datasets demonstrate the superiority of the proposed method in comparison with state-of-the-art deep hashing methods.

## 1. Introduction

Since the rapid advancements of information technology, a vast amount of data has dramatically accumulated in different fields, and we are now living in the century of big data. A major challenge is determining how users can rapidly and precisely search and retrieve the exact data from large-scale databases. Effective and efficient retrieval of data from big datasets has turned into a hot research direction in both industries and academia. As a result of highly efficient performance processing high dimensional data, hashing method has emerged as a solution to this challenge in recent years. The primary goal of hash learning methods is to transform the data item into low-dimensional vectors; a small code containing a series of bits is known as binary hash code [1,2]. Hashing techniques can be categorized into two different groups. Data-independent and data-dependent. The key dissimilarity amongst these methods is that the data-independent method either generated or manually designed the hash function. In contrast, the data-dependent method automatically learns from the data. The most broadly used data-independent hashing methods which perform image retrieval or related tasks are locality-sensitive hashing (LSH) [3] and its variant hashing methods such as Superbit LSH [4], KLSH [5], NLSH [6], and having faster computing hash functions [7]. Using LSH methods, a set of random hyperplanes are generated from a gaussian distribution because a threshold function is applied to the projection results as a projection of the original high dimensional data. The arrival of LSH has enhanced the performance of image retrieval, resulting in a new approach for resolving the issues related to large-scale image retrieval. However, the obtained hash function in data-independent methods shown by LSH is randomly generated or designed. As the number of bits increases, the algorithm’s accuracy increases slowly when it uses a data-independent method like LSH, where the hash function is generated randomly or manually, regardless of the original data distributions. As a result, achieving stable retrieval results in practical applications is not easy. In contrast, the data-dependent method learns the hash function for each code from the data and is practically very significant. In the present days of research and application, data-dependent methods are gaining popularity. Spectral Hashing [8] is one of the classical; data-dependent methods, which aims to use the whole datasets from the training datasets to assemble a entire graph. By using the similarity amongst the data samples as weights on edges. The graph can be viewed as a slice made by each hash function. At least one cut at each edge of the graph meets the requirement that the corresponding sum weight sum of cut edges be the lowest, and the entire, complete graph is split into two portions as evenly as possible. As the classical normalized cut problem in graph theory, the hash code is solved by signing eigenvector that represents the lower eigenvalue of the Laplacian matric. Based on the label information, the data-dependent method can be categorized further into supervised and unsupervised methods. In contrast to unsupervised method, supervised methods strive for semantic similarity rather than similarity into original spac. Unsupervised hashing methods such as Iterative quantization hashing method [9], Scalable graph hashing method [10], discrete graph hashing method [11], and Isotropic hashing method [12] are some examples of unsupervised hashing methods. Supervised hashing methods such as kernel supervised hashing method [13], two-step hashing method [14], supervised discrete hashing [15], Fast supervised hashing method [16,17], and fast discrete supervised hashing method [18], rank based supervised hashing method [19] supervised discrete discriminative hashing [20] and discrete semantic ranking hashing [21]. Unsupervised hashing algorithms cannot achieve adequate retrieval performance in actual applications due to the semantic gap, which refers to the lack of correspondence between the low-level information extracted from an image and the image’s interpretation for a user. How to transform the features computed from raw image data to the high-level representation of semantics carried out by that image. Both unsupervised and supervised hashing methods can be implemented using the quantization-based hashing method (QBH) [22]. A number of fields, including speech recognition and computer vision, have seen outstanding results from deep learning in recent years; many researchers have used deep learning with hashing methods. The first work which has been done in this regard is semantic hashing [23], which used deep learning in their model after widely using the combination of hashing with deep learning [24,25] by the researcher in their work. Such as semi-supervised deep learning hashing [26], convolutional neural network hashing[27], Network-to-Network hashing [28], deep semantic ranking-based hashing [29], deep hashing based on classification and quantization error [30], adaptive similarity hashing [31], deep pairwise supervised hashing [32], and deep discrete supervised hashing [33]. Using the deep learning framework, deep pairwise supervised hashing extracted feature sets and learned hash codes for application and its label. A deep learning model is likely necessary to address complicated learning tasks [34]. Deep neural networks, at present, are the most widely used deep models. Despite the power of the deep neural network, there is still some limitation. According to recent studies, combining supervised information improves hash learning performance. Regardless, as we mentioned earlier, there are still some difficulties and problems in those hashing methods. The first issue is that traditional handcrafted features do not contain complete semantic information. Furthermore, some end-to-end methods consume more memory than other methods. In addition, some methods require much time during the data preparation. In order to obstacle these issues. This study presents a method to retrieve large-scale images using supervised deep hashing, as illustrated in Figure 1. Our proposed model employs CNN’s learning capabilities to extract the features (semantic-similarity) of the input images. The hash function in this framework was built on CNN to learn the nonlinear transformation. Furthermore, it builds a continuous link between binary information and image pixels to retrieve the image. Depending on certain limits on the highest layer of the deep neural network, the proposed method is optimized using the stochastic gradient descent approach and the backpropagation algorithm. The following list summarizes the significant contributions of the proposed method:The deep convolutional neural network-based hash coding approach is introduced and employs multiple nonlinear projections to generate the additional distinctive short binary codes. To extract a rich representation of mid-level information, CNN is applied as the basis of the network. Meanwhile, hash encoding and concurrent learning of the feature representation in back to the back network make the feature of binary code consistent.To maintain the semantic-similarity, a loss function has been designed that naturally pushes the codes of different images away and pulls the codes of similar images together. In the meantime, we minimize the quantization error by providing a uniform distribution of the hash bits.Finally, Extensive experiments have been conducted to test our method against three benchmarks: NUSWIDE, MIRFLICKR25, and MS COCO for retrieval tasks. Comparative results illuminate that our method outperforms state-of-the-art supervised hashing methods. Results of comparative tests reveal that our proposed approach outperforms state-of-the-art supervised hashing.

This article is summarized as follows. In Section 2, we present the proposed method and primary model. Section 3 demonstrates the experimental results and baseline matrices, Section 4 presents the experimental setting and analysis compared with the state of art methods, and Section 5 summarizes the conclusion

## 2. Proposed Methodology

This study utilizes the supervised-based deep hashing method to generate compact binary code. An overview of the proposed method is presented in Figure 1, consisting of three main stages. The network parameter was initialized in the first stage. In the second stage, our method used the images with datasets labeling information to fine-tune the network. At last, with the completion of the training process, binary codes were derived from the network’s outputs based on input images and quantization. The proposed method forced the generated binary code to meet the following requirements to enhance the quality and bit size. (1) Ideally, similar images should be encoded into a binary code that is relatively similar. In contrast, those not similar should be as different as possible. (2) With the evenly distributed hash bits, the quantization error should be reduced from hamming to euclidean space. Our method is further explained as follows.

### 2.1. Network Architecture

There are two main parts of the deep architecture training phase. The first part is the initialization of the network, and the other is the optimization. Due to the excellent image classification performance, the well-known CNN-based initial architecture “GoogleNet” is adopted in our model to extract information as the basic structure of the hash. We initialize the network with a pretrained “GoogleNet” from a Caffe model trained on various large-scale ImageNet datasets. Over one million images are included in this dataset, which is divided into 100 categories. The fully-connected layers are replaced by the last convolutional layer of “Google Net” to force the learning of compact binary codes for hashing tasks. In the second stage, the network is tuned to different dataset criteria for image retrieval using stochastic gradient descent techniques and backpropagation algorithms. The following sections describe the details of the loss function and parameter update.

### 2.2. Loss Function

Suppose we have *Ω* as the space of an initial image. Our primary objective is to learn the mapping from *Ω* to a binary code of K bit: *Ω→*${\left\{+1,-1\right\}}^{K}.$ Such that comparable binary codes are used to represent similar images, whether those images are similar semantically or visually. In this sense, the codes for similar images should be as close as possible, while the codes for different images should be far away. Based on this, the loss function is designed to naturally push the codes of different images away from each other and pull the codes of similar images together. 

We have pair of images *I_1_ I_2_*, and the binary code for each pair of images is represented as *c_1_, c_2_*. The hash code length has been indicated as K. Meanwhile, we ensure that the quantization error is reduced to the minimum, and the hash bits are constantly distributed to allow additional data. According to the definition of the loss associated with this pair of images can be written as follows:(1)$$\begin{array}{l} J=\frac{1}{2\frac{n!}{(n-2)!2!}}\sum_{i=1}^{\frac{n!}{(n-2)!2!}}YD(c_{1},c_{2})+(1-Y)\max(m-D(c_{1},c_{2}),0)\\ +\frac{1}{2n}\sum_{i=1}^{n}\frac{1}{K}\sum_{i=1}^{K}({\left\|c_{1}-v_{1}\right\|}^{2}+{\left\|c_{2}-v_{2}\right\|}^{2})+\frac{1}{2K}\sum_{j=1}^{K}{\left\|\frac{1}{n}\sum_{i=1}^{n}c_{i}(j)-0\right\|}^{2} \end{array}$$
where *Y* indicates the semantic similarity of images, we describe *Y* = 1 if the image features are similar and *Y* = 0 otherwise. The hamming distance amongst the binary vectors is denoted as D(⋅,⋅), and *m* is a margin threshold, which is *m* > 0. As stated in Equation (1), where the first term aims to force the hash code of the same images to be closer and dissimilar images to be far away from each other so that their hamming distance will larger. We can write this part as follows for clarity.
(2)$$J_{1}=\left\{\begin{array}{ll} \;\frac{1}{2}D(c_{1},c_{2}) & Y=1\\ \frac{1}{2}\max(m-D(c_{1},c_{2}),0) & Y=0 \end{array}\right.$$

Such that $c_{i}\in {\left[-1,+1\right]}^{K},i\in \{1,2\}$.

Network entries consist of batches of images. Our network forces each Image and the latter Image to be paired. Hence, if there are *n* images in the batch, then there are *n* pairs will be $C_{n}^{2}=\frac{n!}{2!\left(n-2\right)!}$. Figure 2 shows how the combination is achieved. Using Equation (2) to train the network using a backpropagation algorithm was an excellent idea. Nevertheless, this is challenging because of its indistinguishable properties. A popular technique, *tanh* or *sigmoid*, is used for overcoming this problem, which aims to constrain the output within {·1, +1}. This allows us to relax the integer constraint of the series constraint. However, this type of technique slows down network convergence. Therefore, relax the binary constraint and replace it with {−1,1} with {−1, +1}. Then Equation (2) can be written as: (3)$$J_{1}(c_{1},c_{2})=\left\{\begin{array}{ll} \frac{1}{2}{\left\|c_{1}-c_{2}\right\|}^{2} & Y=1\\ \frac{1}{2}\max(m-{\left\|c_{1}-c_{2}\right\|}^{2},0) & Y=0 \end{array}\right.$$

Such that $c_{i}\in {\left[-1,+1\right]}^{K},i\in \{1,2\}$.

We have used the l_2_ norm in the upper part of the loss function, which aims to compute the distance between the network’s outputs. Consequently, the lower norms produce subgradients that do not account for the information involved in different distance magnitudes when they produce image pairs with different distances.

Using the backpropagation algorithm with the minibatch gradient descent method, for this purpose, the gradient of Equation (3) with respect to $c_{i},\;I\in \left\{1,2\right\}$ needs to be measured as
(4)$$\frac{\partial J_{1}}{\partial c_{i}}={\left(-1\right)}^{i+1}(c_{1}-c_{2})$$
(5)$$\frac{\partial J_{1}}{\partial c_{i}}=\left\{\begin{array}{cc} {\left(-1\right)}^{i}(c_{1}-c_{2}) & {\left\|c_{1}-c_{2}\right\|}^{2}<t\\ 0 & {\left\|c_{1}-c_{2}\right\|}^{2}\geq t \end{array}\right.$$

Such that $c_{i}\in {\left[-1,+1\right]}^{K}$, where i∈{1,2}.

Equation (4) can be used to calculate the gradient when the value of *Y*
*=* 1; alternatively, when the value of *Y* = 0, the gradient can be driven using Equation (5). A binarization step is required since the network’s outputs are real-valued. The second portion of Equation (1) goal is to reduce the error within hamming space, as follows.
(6)$$J_{2}=\frac{1}{K}\sum_{i=1}^{K}{\left\|c_{i}-v_{i}\right\|}^{2}$$

The network’s real-valued outputs are measured in units called *vi*. In order to increase the capacity for information. We advocate a uniform distribution of compact binary codes in the initial part of (1). More information can be transferred if a binary code’s likelihood of −1 or 1 is closer to 50%. This results in a sum of bits that is nearly zero. Therefore, the loss function is as follows.
(7)$$J_{3}=\frac{1}{K}\sum_{j=1}^{K}{\left\|\frac{1}{n}\sum_{i=1}^{n}c_{i}(j)-0\right\|}^{2}$$

The $c_{i}(j)$ represents a *jth* bit of *ith* binary code; the number of binary codes is indicated by *q*. The binary code is given by *n*, where *n* indicates the length. The proposed approach ensures that the network system uses the mentioned loss function to secure image semantic similarity and enhance retrieval performance. Additionally, it reduces quantization error and evenly distributes binary codes. After the network’s training face has been completed, the model generates the q-bit binary code for image testing. We initially feed an image into a network and encode it into a *K*-dimensional real feature vector, as illustrated in Figure 1. Then, regarding the network’s outputs, a basic quantization c=sign(v) results with *K*-bit binary code, as we stated earlier.

## 3. Experiments 

### 3.1. Dataset 

The performance of the proposed method is evaluated on three widely used datasets, and the results are compared with other state-of-the-art methods.

MIRFLICKR-25: This is a smaller version of the MirFlickr25K dataset [35]. The dataset has 25,000 images divided into 24 categories. In our experiments, datasets are classified according to their raw annotations. Each image in the collection is labeled with a 24-dimensional vector corresponding to 24 different object types. We use 10,000 images to train our hash encoding method and another 5000 images to test the hash model for image assignment. Table 1 lists the dimensions of the three datasets and the number of training, test, and labels applied to each dataset.

*NUSWIDE:* Nearly 270,000 image URLs from Flicker are collected in the NUSWIDE dataset [36]. Images are divided into 81 categories, some of which have multiple labels. In our study, we used a subset of the NUSWIDE dataset. The 21most typically utilized labels were used in our process using the previous report. A minimum of 5000 images are associated with each label

*MS COCO:* The current dataset contains 82,783 images from the training set and 40,137 images from the validation set [37]. The dataset generates five sentences per image as ground truth labels based on the 80 most persistent categories. There are 82,081 training images, and some images without categories have been removed from the training set.

### 3.2. Evaluation Matrices

To evaluate experimental performance, we use mean average precision (mAP) to measure the quality of retrieving database images, which has been shown to be discriminative and stable.
(8)$$mAP=\frac{1}{S}\sum_{s=1}^{S}AP(s)$$

Here, the number of related images is denoted as S, and A.P. presents the average precision value of the first N images after each related image is retrieved. Then calculate the average of these values.
(9)$$AP=\frac{\sum_{p=1}^{G}\prod r_{p}.P@N}{G_{r}}$$
where a function ∏(⋅)∈{0,1} denotes an indicator function of $r_{p}>0.\;r_{p}$ corresponds to the similarity to the query image ranked p-th and $G_{r}>0$ represents the number of related images. To calculate it, we can use the average of the similarity of the top N images to the query image.
(10)$$P@N=\frac{1}{p}\sum_{i=1}^{p}r_{i}$$
where P@N is the precision weighted by the similarity level of each image

## 4. Experimental Setting and Result Analysis

Due to remarkable performance in image classification, a famous CNN-based inception architecture, “GoogleNet,” has been adopted in our model for extracting the information as the basic structure to hashing. Our network was initially trained on pre-trained GoogleNet data from the Caffe model, used in various large-scale imageNet datasets. GoogleNet replaces its last convolutional layer with a fully connected layer for the hashing task to enforce compact binary code learning. In the second phase, the network is adjusted to the different data set standards for image retrieval using the stochastic gradient descent technique and backpropagation algorithm. In the next section, we will describe the details of the loss function and parameter updates

### 4.1. Comparison with Other Methods

Table 2 shows mAP values for all baselines that were compared. The results of our method on distinctive data sets with different binary code lengths show that it performs significantly better than each of the competitive baselines. On MIRFlickr25K and MSCOCO NUSWIDE datasets, the performance of our proposed method was enhanced by 11.05, 38.45, and 23.27%, respectively, with the comparison of NNH methods. On the three widely used datasets, the mAP values obtained by our method are 10.02%, 8.91%, and 12.37% higher than those obtained by other methods. It is clear from such improvements that our method is effective.

Figure 3 shows the precision-recall curves for three different datasets with 16-bit hash codes. Besides reporting the mAP and precision curves of the 16-bit hash code on these top retrieved data samples, we also report the efficiency of P@5000 evaluated by various top retrieved data samples, as shown in Figure 4 and Figure 5, respectively. The average performance gaps between PCAH and proposed method retrieval performance on three large datasets are 14.31%, 38.61%, and 23.17%. Comparing the proposed method to DSH achieved an average performance improvement of 10.02%, 8.79%, and 12.287% based on P@5000 values for three datasets. Figure 5 illustrates a consistent outperforming of all other states of art methods on various datasets. The figure shows the precision-recall curve for three different datasets, and the area under the curve represents a significant performance.

Furthermore, we found that the methods also yielded high accuracy at low recall points, which is enough for implementing such a system for image retrieval. Therefore, the results indicate that our proposed technique is significantly better than all baseline approaches evaluated on various datasets in terms of the performance of mAP, P@5000, and P.R. curves. Our proposed method shows excellent performance compared with other states of the art approaches, proving its superior efficiency.

Table 3 illustrates the retrieval performance of the mAP result. During comparisons with other methods on three datasets, the proposed method consistently outperformed the other methods; for example, compared with the competitive method CNNH [38], our method can achieve significant improvement with an average performance of 6.01%, 4.71%, and 5.67% MIRFlickr25K, MSCOCO, and NUSWIDE, respectively. Based on three widely used data sets, we find that the average performance gap is 2.01%, 1.07%, and 2.30%. In Table 4, you will find the P@5000 performance results. On average, the proposed method significantly outperforms CNNH and DNNH [39] when measured in detail, achieving performance improvements of 4.81%, 4.95%, and 4.95%, respectively.

Meanwhile, compared with DNNH, our method improves average performance with 2.32%, 2.07%, and 3.15% on MIRFlickr25K, MSCOCO, and NUSWIDE, respectively. Although our method performed better in mAP and P@5000 than SDH, the best deep hashing algorithm uses a binary quantization function. In Figure 6, we have also included some retrieval findings for the top 11 returned samples based on hamming ranking on the NUSWIDE, MIRFlickr25K. We can see if the recommended strategy produces better outcomes than the other options. The NUSWIDE dataset illustrates the considerable performance of our technique, precisely the suggested method, which achieves superior retrieval results than other state-of-the-art methods. The proposed approach could better retain the image pair’s similarity while producing a discriminative hash code.

Impact of parameter: A dimension K of the feature space is considered with a single parameter in the proposed method. Using multi-dimensional linear search, we analyze the impact of the different datasets on ($2^{1},2^{2},2^{3},2^{4},2^{5}$). In particular, we fix the code length to 60 bits as log_2_K, which is the least common multiplier to 60. In Figure 7, the mAP results and P@5000 for image retrieval are based on the retrieval method system. For the MIRFlickr25K dataset, different K settings result in only a very slight performance impact. In contrast, it decreases when K is set to 2 on MSCOO. The setting of K for MIRFlickr25K and MSCOCO can be determined from Figure 7, and for NUSWIDE, our setting can be determined from Figure 7.

### 4.2. Convergence Analysis and Time Complexity

The proposed method convergence and time complexity has been evaluated through some experiments. Time complexity comparison with other state-of-the-art hashing methods has been illustrated in Table 5. The convergence was evaluated using the loss. A comparison of three datasets based on the proposed method shows changes between the three datasets, as shown in Figure 8. The value of loss becomes smaller and more stable as the number of iterations increases. During training, the proposed method appears to reach convergence quickly, significantly reducing training time.

## 5. Conclusions

A CNN based supervised deep hashing method is implemented in this article that aims to achieve high-quality bit binary code with efficient performance for image retrieval. Two different perspectives assessed the proposed method. In a single aspect, the simultaneous hash coding learning of feature representation makes the hash code fit with the features. Furthermore, the designed loss function maintains the original space’s similarity by compelling the binary codes. While optimizing the quantization, the hash bits have been allocated constantly. Experiments on three extensively used standard datasets have been performed to pour the proposed system exceeding the specific state-of-art algorithms.

## Figures and Tables

**Figure 1 entropy-24-01425-f001:**
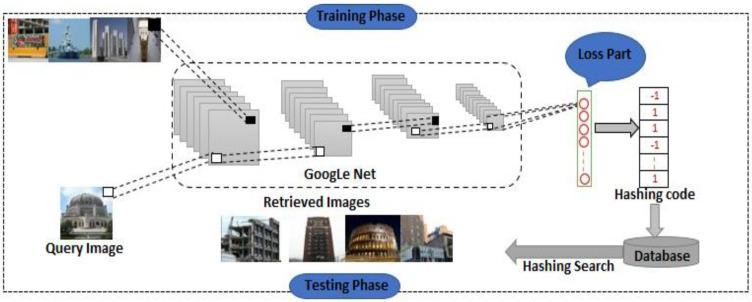
Basic structure of our model. The primary network is GoogleNet, and our hash layer replaces the last classification layer of the GoogleNet convolutional layer. The hash bits of the layer define their unit number. An image batch is fed to the network during the training as input. By coding similar images in the same binary patterns, we force them to be coded the same way and vice versa, in addition to minimizing quantization errors and ensuring that hash bits are evenly distributed. The trained network is tested by inputting a new image, and this is quantized into [+1, −1].

**Figure 2 entropy-24-01425-f002:**
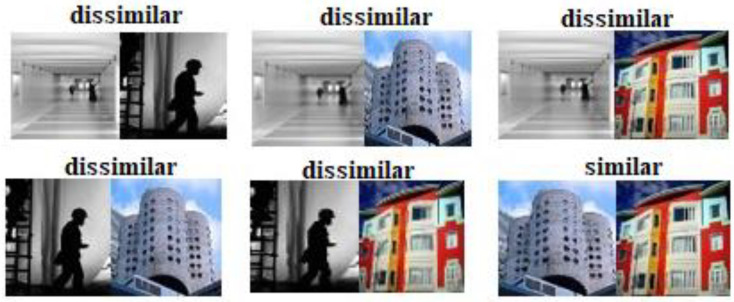
Assume a batch consisting of four images as illustrated. The loss calculation is always done as a pair for each and its last image within a batch. The total number of pairs is denoted as $C_{n}^{2}=\frac{n!}{2!(n-2)!}=6$.

**Figure 3 entropy-24-01425-f003:**
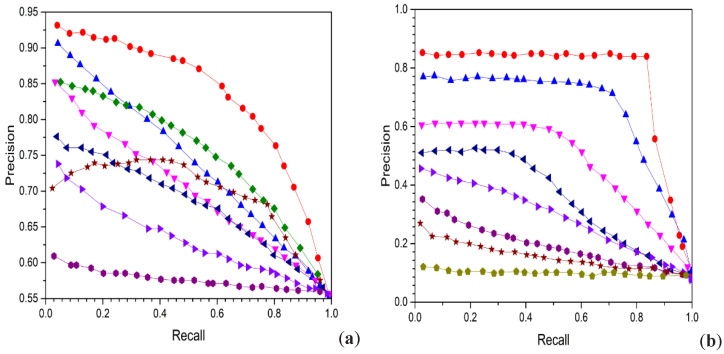
Precision recall curve results with 16-bit hash code for various methods. (**a**) MIRFLICKR25K, (**b**) NUSWIDE, (**c**) MSCOCO.

**Figure 4 entropy-24-01425-f004:**
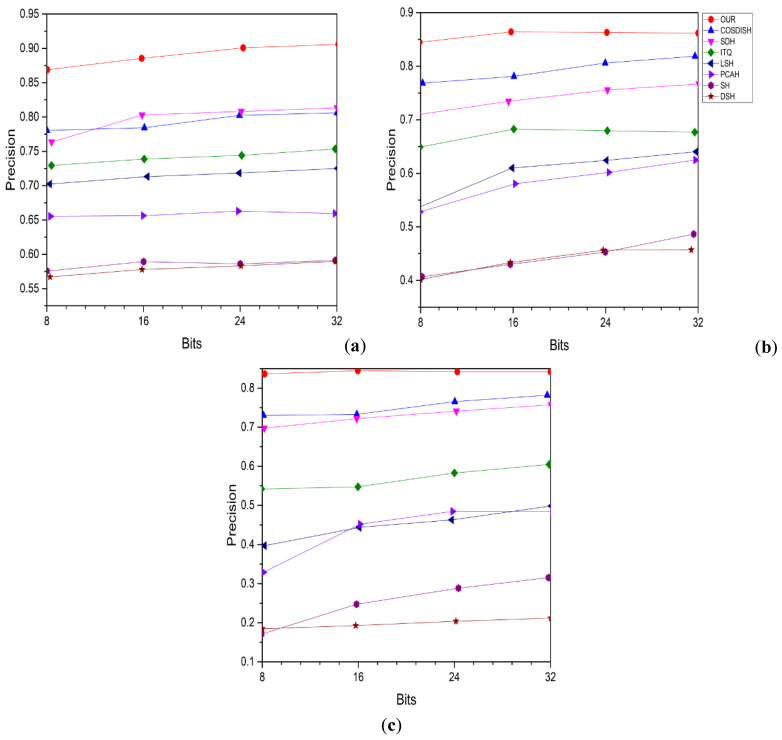
PRC results of competitive methods with 16 bits hash code on three large datasets. (**a**) MirFlickr-25K, (**b**) MS-COCO, (**c**) NUS-WIDE.

**Figure 5 entropy-24-01425-f005:**
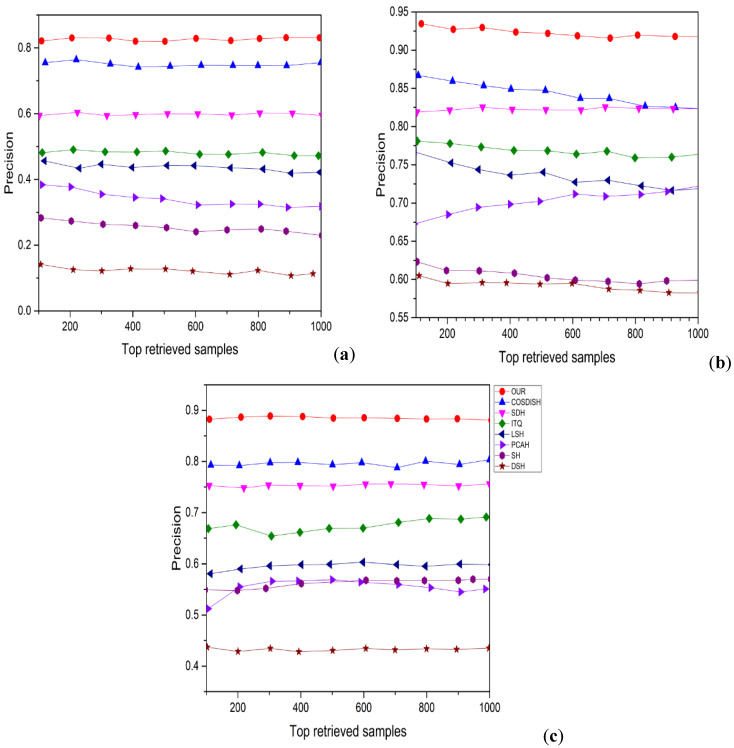
P.R. result based on top three returned datasets with 16 bits hash code. (**a**) MIRFlickr25K, (**b**) NUSWIDE, (**c**) MSCOCO.

**Figure 6 entropy-24-01425-f006:**
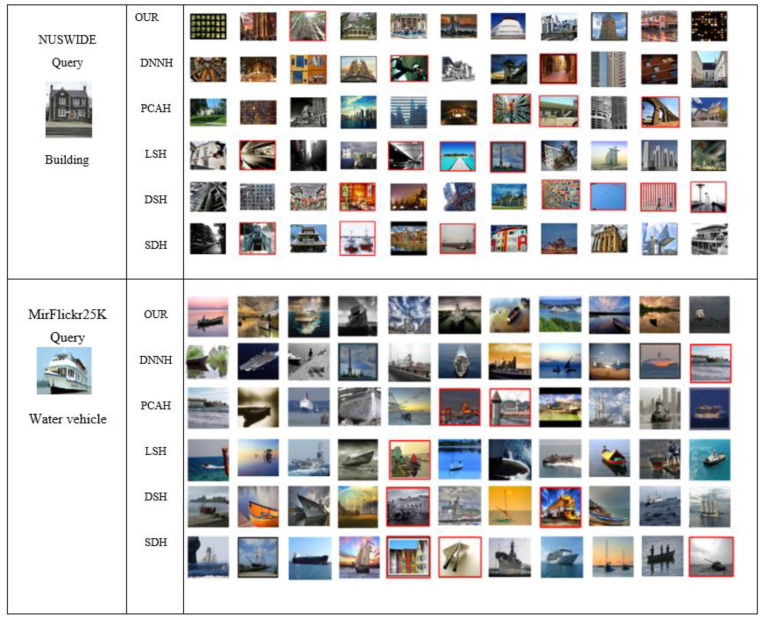
Data retrieval results for MirFlickr25K and NUSWIDE datasets. Query image displayed on the left side of the table. On the right are search results for different competing methods, listed in the middle column. 32-bit codes were used in all cases, and Hamming’s ranking determined the top 11 results. Red rectangles represent incorrect results.

**Figure 7 entropy-24-01425-f007:**
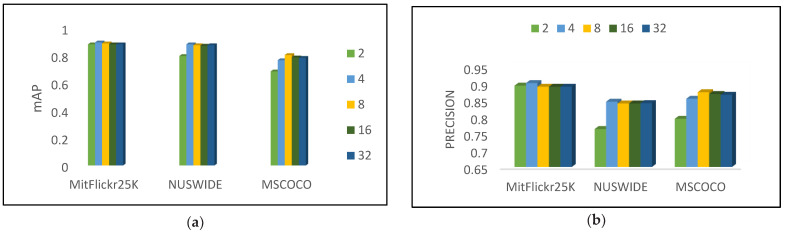
(**a**) mAP results and (**b**) P@5K result on three datasets for different K’s for 60 bits-hash code.

**Figure 8 entropy-24-01425-f008:**
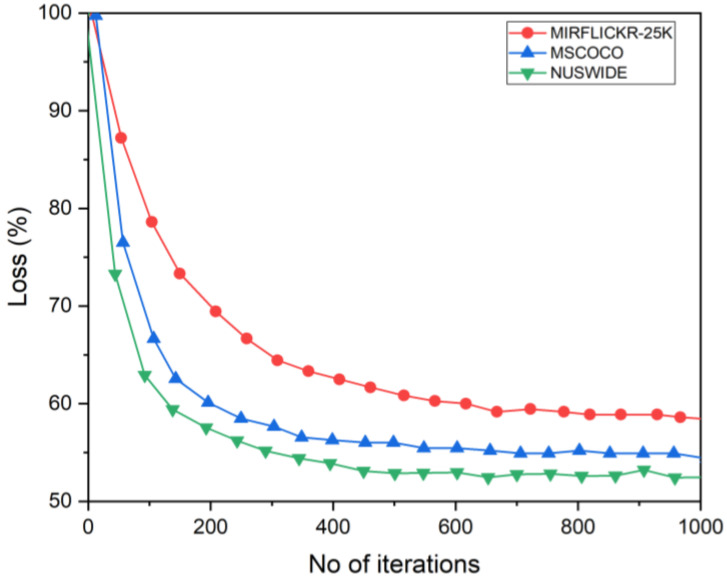
The convergence reduction results of three datasets with 1000 iterations. The code length is 16 bits.

**Table 1 entropy-24-01425-t001:** Statistics of the experimental dataset.

Dataset	NUSWIDE	MIRFLICKR-25K	MS COCO
Dataset size	195,834	20,015	85,000
Training set	10,500	10,000	10,000
Query set	2100	2000	5000
No of Labels	21	24	80

**Table 2 entropy-24-01425-t002:** Results of mAP with different lengths of code on three datasets. Bold to indicate the values for proposed method.

Models	Datasets											
	MIRFLICKR-25K				MSCOCO				NUSWIDE			
	8 Bits	16 Bits	12 Bits	32 Bits	8 Bits	16 Bits	12 Bits	32 Bits	8 Bits	16 Bits	12 Bits	32 Bits
**OUR**	**0.8701**	**0.8810**	**0.8740**	**0.8771**	**0.8712**	**0.8696**	**0.8749**	**0.880**	**0.7602**	**0.7795**	**0.7887**	**0.7989**
COSDISH	0.7453	0.7748	0.7909	0.7970	0.4253	0.4909	0.5151	0.5395	0.5339	0.5570	0.5732	0.5822
SDH	0.7268	0.7292	0.73477	0.7416	0.3371	0.4779	0.5142	0.5200	0.5221	0.5394	0.5415	0.4588
ITQ	0.6448	0.6472	0.6515	0.6517	0.1767	0.1839	0.1858	0.1902	0.4725	0.4835	0.4880	0.4943
LSH	0.5821	0.5845	0.5826	0.5863	0.1087	0.1137	0.1172	0.1331	0.3845	0.4047	0.4090	0.4090
PCAH	0.6843	0.6968	0.7001	0.7035	0.3096	0.3599	0.3766	0.3892	0.4985	0.5084	0.5160	0.5236
SH	0.7498	0.7622	0.7745	0.7670	0.1580	0.5959	0.6329	0.6358	0.6472	0.6598	0.6747	0.6784
DSH	0.7127	0.7291	0.7394	0.7353	0.7470	0.7623	0.7773	0.8019	0.6160	0.6370	0.6397	0.6384

**Table 3 entropy-24-01425-t003:** Three mAP results for proposed and other methods using multiple datasets with different code lengths.

Models	Datasets											
	MIRFLICKR-25K				MSCOCO				NUSWIDE			
	8 Bits	12 Bits	16 Bits	32 Bits	8 Bits	12 Bits	16 Bits	32 Bits	8 Bits	12 Bits	16 Bits	32 Bits
OUR	0.8701	0.8810	0.8740	0.8712	0.8712	0.8696	0.8749	0.8803	0.7602	0.7795	0.7887	0.7989
CNNH	0.8002	0.5210	0.8321	0.3247	0.8075	0.8193	0.8345	0.8387	0.6993	0.7190	0.7285	0.7383
DNNH	0.8260	0.8510	0.8675	0.8693	0.8395	0.8487	0.8591	0.8596	0.7451	0.7791	0.7869	0.7901

**Table 4 entropy-24-01425-t004:** A comparison results of proposed and other methods of P@5000 with a different hash length of hash code on three datasets.

Models	Datasets											
	MIRFLICKR-25K				MSCOCO				NUSWIDE			
	8 Bits	12 Bits	16 Bits	32 Bits	8 Bits	12 Bits	16 Bits	32 Bits	8 Bits	12 Bits	16 Bits	32 Bits
OUR	0.8901	0.8835	0.8792	0.8875	0.8186	0.8177	0.8182	0.8155	0.8252	0.8468	0.8493	0.8500
CNNH	0.8085	0.8471	0.8467	0.8481	0.7951	0.7831	0.7835	0.7761	0.8003	0.8063	0.8115	0.8160
DNNH	0.8310	0.8645	0.8765	0.8730	0.8024	0.8432	0.8362	0.8175	0.8300	0.8434	0.8520	0.8563

**Table 5 entropy-24-01425-t005:** Comparisons of time complexity of different hashing methods.

Methods	Time Complexity
SDH	$O(m^{2}n+(s+1)kn)$
DSH	$O(nmkN_{B}N+k^{2}nN)$
LSH	$O(n^{3})$
PCAH	$O(mn+(s+1)kn+2nk^{2}N+nkN\log_{2}n)$
DNNH	$O(nmkN_{G}N+nkN\log_{2}n)$
OUR	O(2ndkN+ndm)

## Data Availability

The datasets used and/or analyzed during the current study are available in https://press.liacs.nl/mirflickr/mirdownload.html (accessed on 1 November 2021); https://lms.comp.nus.edu.sg/wp-content/uploads/2009/research/nuswide/NUS-WIDE.html (accessed on 10 October 2021), https://cocodataset.org/#home (accessed on 12 October 2021).

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
