# Peer review of "An Efficient Supervised Deep Hashing Method for Image Retrieval"

_entropy, 2022, doi:10.3390/e24101425_

Round 1

Reviewer 1 Report

The work presented in the paper is interesting and methodologically sound. Anyway I've three minor requests for the authors.

1. In the introduction, the authors say that many other researchers have used hashing methods that can also be utilized with CNN, but you don't furnish any references about this; the authors should insert some references.

2. The authors should explain throughout the text what do they mean with "semantic gap".

3. The authirs should rewrite the formulas of the loss function they used; there are typing errors in equations 4 and 5, and the total number of pairs (6)  presented before equation 1 must be explained.

Author Response

Dear Prof. Mr. Alexandru-Marius Barză

Assistant Editor and Reviewers,

Thank you for giving us the opportunity to submit a revised draft of the manuscript "An efficient supervised deep hashing method for image retrieval" in the journal of "Entropy." We appreciate the time and effort you and the reviewers dedicated to providing feedback on our manuscript and are grateful for the insightful comments and valuable improvements to our paper. We have updated the original manuscript and submitted a revised version according to your suggestions.

All the comments raised by the associate editor and the reviewers have been addressed in this review response report. To enhance the legibility of this report, all the Reviewer's comments are typeset in italic font, and our responses are printed in plain font.

We are thankful to the reviewers and editors again for reviewing this paper. We will be delighted to continue to improve this work if there are more suggestions and comments from the reviewers and editor.

Yours Sincerely, Abid Hussain

Reviewer 2 Report

General Comments:

The paper deals with large-scale image retrieval and presents a deep-learning approach for hashing using CNN, with multiple nonlinear projections to transform the image into a binary code. The topic is important in computer vision and big data processing, and the presented hashing approach is interesting. However, lack of clarity and insufficient reasoning are big issues in this paper.

Specific Comments:

1.    The Abstract should be more expressive and quantitative regarding the contributions of this work versus existing hashing methods.

2.    There should be a clear link with entropy to justify publication in this journal.

3.    Section 3:

a)   The proposed system has suddenly been introduced without sufficient analysis or details.

b)   It is unclear whether the proposed approach is dependent on the databases under test.

c)    The length K of the binary code should be explained.

d)   It is unclear how the loss function in Sub-Section 3.2 can handle similarity between images. Clear analysis is required.

e)   The variables and parameters used to define the loss functions in Sub-Section 3.2 should all be well-defined and well-explained.

f)      In Sub-Section 2.1, the Authors criticized the tanh activation, while they use it in Sub-Section 3.2. Please clarify this point and its relation to the vanishing gradient issues.

g)   The norm in Equation (3) should be defined then justified.

4.    Section 5:

a)   All performance metrics should be defined in a clear mathematical form.

b)   There should be a comparison showing reduced convergence speed as promised in Sub-Section 3.2. Comparisons with existing methods should include the time-complexity.

5.    Language Usage: The paper is in need for a moderate language revision, including punctuation and capitalization of letters.

Author Response

(The authors gave the same response as above.)

Round 2

Reviewer 2 Report

The Authors have addressed the Reviewer’s concerns sufficiently.

The current version is suitable for publication.